# Quantifying the factors limiting rate performance in battery electrodes

Ruiyuan Tian[1,2,5], Sang-Hoon Park [1,3,5], Paul J. King[4], Graeme Cunningham[1,3], João Coelho [1,3], Valeria Nicolosi[1,3] & Jonathan N. Coleman [1,2]

One weakness of batteries is the rapid falloff in charge-storage capacity with increasing charge/discharge rate. Rate performance is related to the timescales associated with charge/ionic motion in both electrode and electrolyte. However, no general fittable model exists to link capacity-rate data to electrode/electrolyte properties. Here we demonstrate an equation which can fit capacity versus rate data, outputting three parameters which fully describe rate performance. Most important is the characteristic time associated with charge/discharge which can be linked by a second equation to physical electrode/electrolyte parameters via various rate-limiting processes. We fit these equations to ~200 data sets, deriving parameters such as diffusion coefficients or electrolyte conductivities. It is possible to show which rate-limiting processes are dominant in a given situation, facilitating rational design and cell optimisation. In addition, this model predicts the upper speed limit for lithium/sodium ion batteries, yielding a value that is consistent with the fastest electrodes in the literature.

[1] CRANN and AMBER research centers, Trinity College Dublin, Dublin, Dublin 2, Ireland. [2] School of Physics, Trinity College Dublin, Dublin, Dublin 2, Ireland. [3] School of Chemistry, Trinity College Dublin, Dublin, Dublin 2, Ireland. [4] Efficient Energy Transfer Department, Bell Labs Research, Nokia, Blanchardstown Business & Technology Park, Snugborough Road, Fingal, Dublin 15, Ireland. [5]These authors contributed equally: Ruiyuan Tian, Sang-Hoon Park. Correspondence and requests for materials should be addressed to J.N.C. (email: colemaj@tcd.ie)

Rechargeable batteries that utilise lithium-ion or sodium-ion chemistry are important for applications including electric vehicles, portable electronics, and grid-scale energy storage systems[1,2]. While electrode design and the development of high capacity materials are relatively advanced, high-rate (power) performance still needs to be improved for a range of applications[3]. In particular, high rate performance is critical for rapid charging and high power delivery[4].

Rate performance in batteries is limited because, above some threshold charge or discharge rate, $R_T$, the maximum achievable capacity begins to fall off with increasing rate. This limits the amount of energy a battery can deliver at high power, or store when charged rapidly. Attempts to solve this problem have involved targeting the electrode[5–8], the electrolyte[9], and the separator[10] with the aim of increasing $R_T$ and reducing the rate of capacity falloff above $R_T$.

The factors effecting high-rate capacity are well known. For example, the rate performance can be improved by decreasing active particle size[11–13], and electrode thickness[14–17], or by increasing solid-state diffusivity[11], conductor content[7,16,18], or electrode porosity[16,19], as well as by optimising electrolyte concentration[14,16] and viscosity[16].

Based on such information, it is accepted that rate performance is limited by: electronic transport in electrodes; ion transport both in bulk electrolyte and electrolyte-filled pores; solid-state diffusion of ions in the active materials and electrochemical reactions at the electrode/electrolyte interface[12,20–22]. One would expect that speeding up any of these processes would improve rate performance.

However, in practice, it is difficult to quantitatively link the observed rate performance to the factors given above. The most commonly reported experimental rate performance data are capacity versus rate curves. Ideally, the experimentalist would be able to fit his/her capacity-rate data to an analytic model which quantitatively includes the influence of the parameters above (i.e. electrode thickness, porosity, particle size, etc.). However, to the best of our knowledge, comprehensive, fittable, analytic models are not available.

A number of theoretical models which describe Li-ion batteries have been reported[23]. Probably most relevant are the electro-chemical models[20,24–26], based on concentrated solution theory[27,28]. Such models provide a comprehensive description of cell operation and match well to experimental data[14]. However, these models involve the numerical solution of a number of coupled differential equations and require knowledge of a large number of numerical parameters. While simplified models have been proposed, they only apply in specific circumstances[22]. As a result, these models are not widely used for fitting purposes. Alternatively, a number of fittable, analytical, physical models are available but only describe the high-rate region[24,29]. Due to these limitations, a number of empirical equations to fit capacity versus rate data have been proposed. However, all are limited in that they only describe a single rate limiting mechanism, generally diffusion[2,30,31].

To address these issues, we have developed a semi-empirical equation which accurately describes the rate dependence of electrode capacity in terms of electrode properties, via the char-acteristic time associated with charge/discharge. Importantly, we derive a simple expression for this characteristic time, which includes the mechanistic factors described above. Together, these equations accurately describe a wide range of data extracted from the literature.

## Results

**Model development**. This work was inspired by recent work on rate limitations in electrically limited supercapacitors[32,33], which describes the dependence of specific capacitance, $C/M$, on scan rate, $v$:[32]

$$\frac{C}{M} = C_M\left[1 - \frac{v\tau_{SC}}{\Delta V}\left(1 - e^{-\Delta V/v\tau_{SC}}\right)\right] \quad (1)$$

where $C_M$ is the capacitance at low rate, $\Delta V$ is the voltage window and $\tau_{SC}$ is the RC time constant associated with charging/discharging the supercapacitor. Unlike diffusion-limited supercapacitors where the high-rate capacitance scales with $v^{-1/2}$, Eq. (1) predicts resistance-limited supercapacitors to show high-rate scaling of $C \propto v^{-1}$, as reported previously[33]. We believe that this equation can be modified empirically to describe rate effects in battery electrodes.

The simplest way to empirically generalise Eq. (1) would be to replacing capacitance, $C$, with capacity, $Q$, and substitute $v/\Delta V$ by a fractional charge/discharge rate, $R$ (this paper will follow the convention that $C$ represents capacitance while $Q$ represents capacity). This will result in an equation that gives constant capacity at low rate but $Q \propto R^{-1}$ at high rate. However, diffusion-limited battery electrodes often display capacities which scale as $Q \propto R^{-1/2}$ at high rate[24]. To facilitate this, we empirically modify the equation slightly so that at high rates, it is consistent with $Q \propto R^{-n}$, where $n$ is a constant:

$$\frac{Q}{M} = Q_M\left[1 - (R\tau)^n\left(1 - e^{-(R\tau)^{-n}}\right)\right] \quad (2)$$

Here $Q/M$ is the measured, rate-dependent specific capacity (i.e. normalised to electrode mass), $Q_M$ is the low-rate specific capacity and $\tau$ is the characteristic time associated with charge/discharge. Although we have written Eq. (2) in terms of specific capacity, it could also represent areal capacity, volumetric capacity, etc., so long as $Q/M$ is replaced by the relevant measured parameter (e.g. $Q/A$ or $Q/V$) while $Q_M$ is replaced by the low-rate value of that parameter (e.g. $Q_A$ or $Q_V$). Although this equation is semi-empirical, it has the right form to describe rate behaviour in batteries while the parameters, particularly $\tau$, are physically relevant.

To demonstrate that Eq. (2) has the appropriate properties, in Fig. 1 we use it to generate plots of $Q/M$ versus $R$ for different values of $Q_M$, $\tau$ and $n$. In all cases, we observe the characteristic plateau at low rate followed by a power-law decay at high rate. These graphs also make clear the role of $Q_M$, $\tau$ and $n$. $Q_M$ reflects the low-rate, intrinsic behaviour and is a measure of the maximum achievable charge storage. Taylor-expanding the exponential in Eq. (2) (retaining the first three terms) gives the high-rate behaviour:

$$\left(\frac{Q}{M}\right)_{\text{high }R} \approx \frac{Q_M}{2(R\tau)^n} \quad (3)$$

confirming a power-law decay with exponent $n$, a parameter which should depend on the rate-limiting mechanisms, with diffusion-limited electrodes displaying $n = 1/2$. Alternatively, by analogy with supercapacitors, other values of $n$ may occur, e.g. $n = 1$ for resistance-limited behaviour[32].

Most importantly, $\tau$ is a measure of $R_T$, the rate marking the transition from flat, low-rate behaviour to high-rate, power-law decay (transition occurs roughly at $R_T = (1/2)^{1/n}/\tau$). This means $\tau$ is the critical factor determining rate performance. As a result, we would expect $\tau$ to be related to intrinsic physical properties of the electrode/electrolyte system.

Before fitting data, the rate must be carefully defined. Most papers use specific current density, $I/M$, or the C-rate. However, here we define rate as

$$R = \frac{I/M}{(Q/M)_E} \quad (4)$$

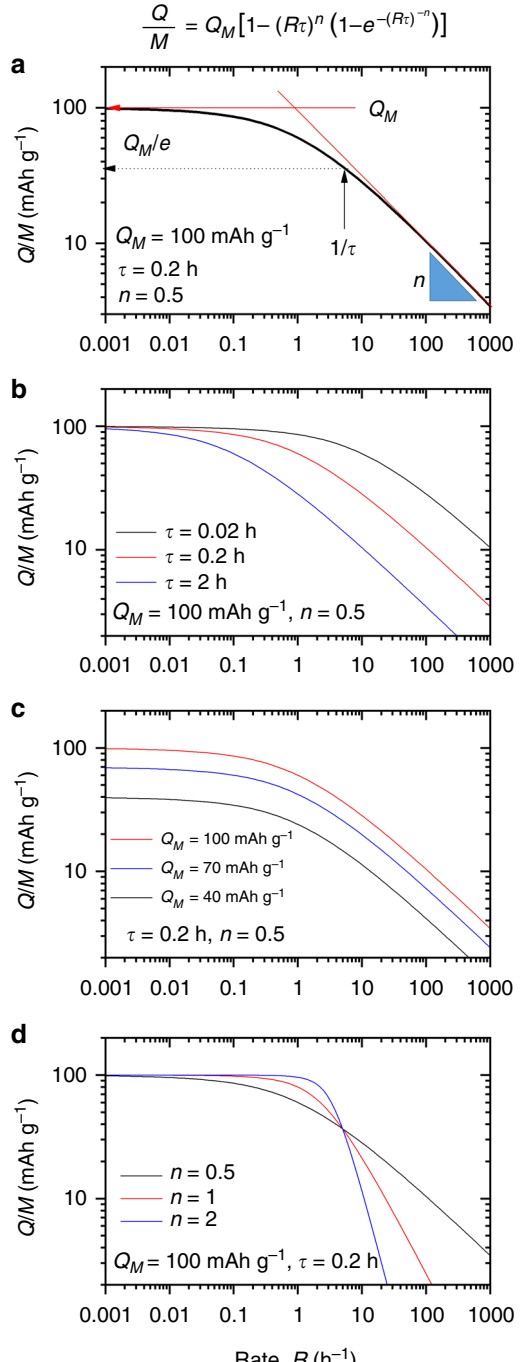

**Fig. 1** Understanding the effect of the parameters defining the model. **a** Specific capacity plotted versus rate using Eq. (2) (also given above panel **a**) using the parameters given in the panel. The physical significance of each parameter is indicated: $Q_M$ represents the low-rate limit of $Q/M$, $n$ is the exponent describing the fall-off of $Q/M$ at high rate and $\tau$ is the characteristic time. The inverse of $\tau$ represents the rate at which $Q/M$ has fallen by $1/e$ compared to its low-rate value. **b**–**d** Plotting Eq. (2) while separately varying $\tau$ (**b**), $Q_M$ (**c**) and $n$ (**d**)

**Fitting literature data**. We extracted capacity versus rate data from a large number of papers (>200 rate-dependent data sets from >50 publications), in all cases, converting current or C-rate to $R$. We divided the data into three cohorts: I, standard lithium ion electrodes[7,16,17,34–51]; II, standard sodium ion electrodes[52–66]; and III, data from studies which systematically varied the content of conductive additive[7,18,19,65,67–73]. Then, we fitted each capacity-rate data set to Eq. (2) (see Fig. 2a and Supplementary Figs. 1−41 for examples), finding very good agreement in all cases (~95% of fits yield $R^2 > 0.99$). From each fit, we extracted values for $Q_M$, $n$ and $\tau$. Because of the broad spectrum of materials studied, the obtained values of $Q_M$ spanned a wide range. As we focus on rate effects, we will not discuss $Q_M$, only refer to these values when necessary.

Shown in Fig. 2b are the extracted values of $n$ and $\tau$ for cohorts I and II. It is clear from this panel that $n$ is not limited to values of 0.5, as would be expected for diffusion-limited systems but varies from ~0.25 to 2.0. In addition, $\tau$ varies over a wide range from <1 s to >1 h.

It is well known that rate performance tends to degrade as the electrode thickness (or mass loading) is increased[17]. Thus, $\tau$ should depend on the electrode thickness, $L_E$, which turns out to be the case (Fig. 2c). Surprisingly, this data shows that for a given $L_E$, sodium ion batteries are no slower than lithium ion batteries, contrary to general perceptions[74]. Interestingly, over the entire data set, $\tau$ scales roughly as $L_E^2$ (solid line). From this scaling, we define a parameter, $\Theta$, which we denote the transport coefficient: $\Theta = L_E^2/\tau$, such that electrodes with higher $\Theta$ will have better rate performance. The frequency of occurrence of $\Theta$ for the samples from cohorts I and II is plotted as a histogram in Fig. 2d. This shows a well-defined distribution with $\Theta$ varying from $10^{-13}$ to $10^{-9}$ m$^2$ s$^{-1}$. As we will show below, $\Theta$ is the natural parameter to describe rate performance in electrodes. In addition, we will show that the upper end of the $\Theta$ distribution represents the ultimate speed limit ($\Theta_{max}$) in lithium/sodium-ion battery electrodes.

Although the $L_E^2$-scaling observed in Fig. 2c seems to suggest that battery electrodes are predominantly limited by diffusion of cations within the electrode, such a conclusion would be incorrect, as we will demonstrate. To see this, we first examine the exponent, $n$.

This parameter is plotted versus $L_E$ in Fig. 2e and displays only very weak thickness dependence. More interesting is the histogram showing the frequency of occurrence of $n$ values in cohorts I and II (Fig. 2f). This clearly shows that most samples do not display $n = 0.5$ as would be expected for purely diffusion-limited systems. In fact, we can identify weak peaks for $n = 0.5$ and $n = 1$ with most of the data lying in between. For supercapacitors, $n = 1$ indicates electrical limitations[32,33]. If this also applies to batteries, Fig. 2 suggests most reported electrodes to be governed by a combination of diffusion and electrical limitations. Interestingly, a small number of data sets are consistent with $n > 1$, indicating a rate-limiting mechanism which is even more severe than electrical limitations. We note that the highest values of $n$ are associated with Si-based electrodes where unwanted electrochemical effects, such as alloying, Li-plating, or continuous SEI formation, caused by particle pulverisation, may affect lithium storage kinetics[75]. In addition, it is unclear why some data points are consistent with $n < 0.5$, although this may represent a fitting error associated with datasets showing small capacity falloffs at higher rate.

**Varying conductive additive content**. The contribution of both diffusion and electrical limitations becomes clear by analysing cohort III of literature data (papers varying conducting additive content). Shown in Fig. 3a are specific capacity versus rate data

where $(Q/M)_E$ represents the experimentally measured specific capacity (at a given current). This contrasts with the usual definition of C-rate $= (I/M)/(Q/M)_{Th}$, where $(Q/M)_{Th}$ is the theoretical specific capacity. We chose this definition because $1/R$ is then the measured charge/discharge time, suggesting that $\tau$-values extracted from fits will have a physical significance.

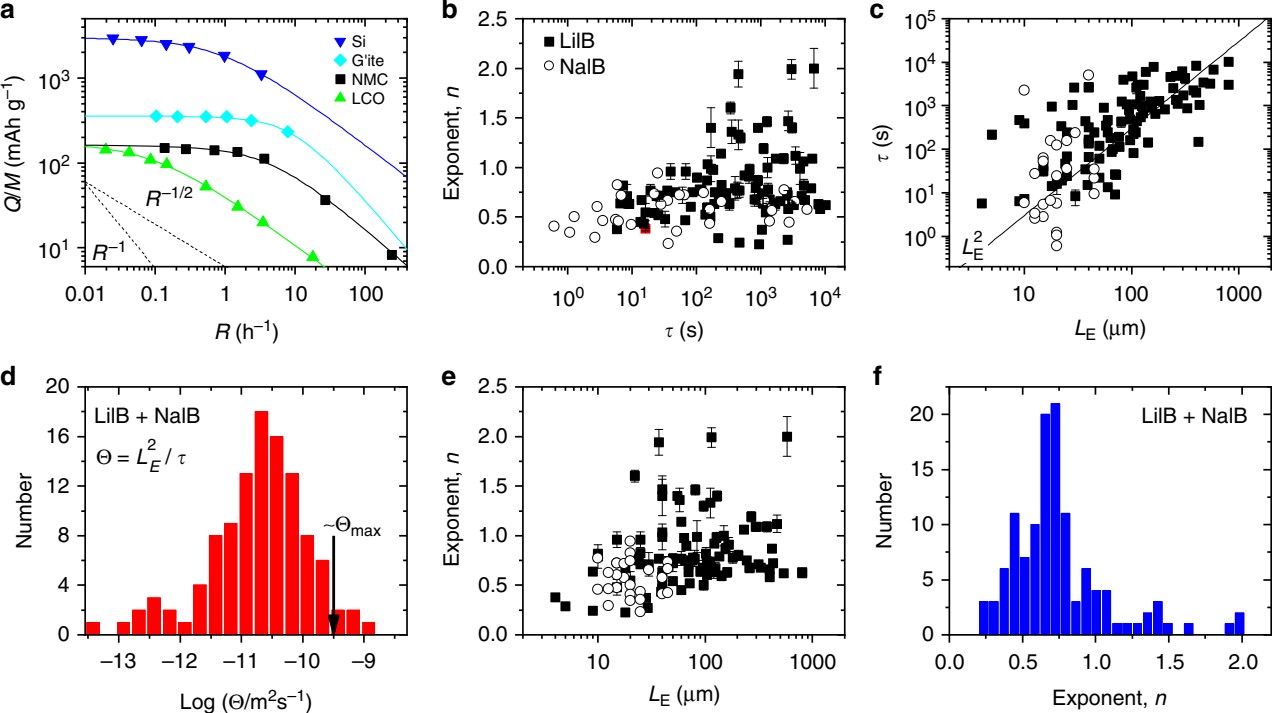

**Fig. 2** Overview of literature data analysed using Eq. (2). **a** Four examples of specific capacity (Q/M) versus rate data taken from the literature. These data all represent lithium ion half cells with examples of both cathodes and anodes. The cathode materials are nickel manganese cobalt oxide (NMC, ref. [39]) and lithium cobalt oxide (LCO, ref. [34]) while the anode materials are silicon (Si, ref. [43]) and graphite (G'ite, ref. [51]). In each case the solid lines represent fits to Eq. (2), while the dashed lines illustrate $R^{-1}$ and $R^{-1/2}$ behaviour. **b** Eq. (2) was used to analyse 122 capacity-rate data sets from 42 papers describing both lithium ion (LiIB) and sodium ion (NaIB) half cells. The resultant $n$ and $\tau$ data are plotted as a map in (**b**) (this panel does not include work which varies the content of conductive additive). **c** Characteristic time, $\tau$, plotted versus electrode thickness, $L_E$ for NaIBs and LiIBs. The line illustrates $L_E^2$ behaviour. **d** Histogram (N = 122) showing frequency of occurrence of $\Theta = L_E^2/\tau$ for NaIBs and LiIBs (log scale). The arrow shows the predicted maximal value of $\Theta$. **e** Exponent, $n$, plotted versus electrode thickness, $L_E$, for NaIBs and LiIBs. **f** Histogram (N = 122) showing frequency of occurrence of $n$ for NaIBs and LiIBs

for anodes of GaS nanosheets mixed with carbon nanotubes at different mass fractions, $M_f$ (ref. [7]). A clear improvement in rate performance can be seen as $M_f$ and hence the electrode conductivity, increases, indicating changes in $\tau$ and $n$. We fitted data extracted from a number of papers[7,18,19,65,67–73] to Eq. (2) and plotted $\tau$ and $n$ versus $M_f$ in Fig. 3b and c. These data indicate a systematic drop in both $\tau$ and $n$ with increasing electrode conductivity.

Figure 3b shows $\tau$ to fall significantly with $M_f$ for all data sets, with some samples showing a thousand-fold reduction. Such behaviour is not consistent with diffusion effects solely limiting rate performance. We interpret the data as follows: at low $M_f$, the electrode conductivity is low and the rate performance is limited by the electrode resistance. As $M_f$ increases, so does the conductivity, reducing the electrical limitations and shifting the rate-limiting factor toward diffusion. This is consistent with the fact that, for a number of systems we see $\tau$ saturating at high $M_f$, indicating that rate limitations associated with electron transport have been removed. We emphasise that it is the out-of-plane conductivity which is important in battery electrodes because it describes charge transport between current collector and ion storage sites[33]. This is important as nanostructured electrodes can be highly anisotropic with out-of-plane conductivities much smaller[33] than the typically reported in-plane conductivities[8,18].

Just as interesting is the data for $n$ versus $M_f$, shown in Fig. 3c. For all data sets, $n$ transitions from $n \sim 1$ at very low $M_f$ to $n \sim 0.5$, or even lower, at high $M_f$. This is consistent with $n = 1$ representing resistance-limited and $n = 0.5$ representing diffusion-limited behaviour as is the case for supercapacitors[33]. Because, electrodes become predominately diffusion limited at

high $M_f$, the values of $n$ tend to be lower in cohort III compared to cohort I and II, especially at high $M_f$, as shown in Fig. 3d.

**The relationship between τ and physical properties.** This data strongly suggests most battery electrodes to display a combination of resistance and diffusion limitations. This can be most easily modelled considering the characteristic time associated with charge/discharge, $\tau$. The data outlined above implies that $\tau$ has both resistance and diffusive contributions. In addition, we must include the effects of the kinetics of the electrochemical reaction at the electrode/electrolyte interface. This can be done via the characteristic time associated with the reaction, $t_c$, which can be calculated via the Butler–Volmer equation[20], and can range from ~0.1 to >100 s ref. [20].

Then, $\tau$ is the sum of the three contributing factors:

$$\tau = \tau_{\text{Electrical}} + \tau_{\text{Diffusive}} + t_c \qquad (5a)$$

It is likely that the diffusive component is just the sum of diffusion times associated with *cation* transport in the electrolyte, both within the separator (coefficient $D_S$) and the electrolyte-filled pores within the electrode (coefficient $D_P$), as well as in the solid active material (coefficient $D_{AM}$)[20]. These times can be estimated using $L = \sqrt{Dt}$ such that

$$\tau_{\text{Diffusive}} = \frac{L_E^2}{D_P} + \frac{L_S^2}{D_S} + \frac{L_{AM}^2}{D_{AM}} \qquad (5b)$$

where $L_E$, $L_S$ and $L_{AM}$ are the electrode thickness, separator thickness, and the length scale associated with active material particles, respectively. $L_{AM}$ depends on material geometry: for a

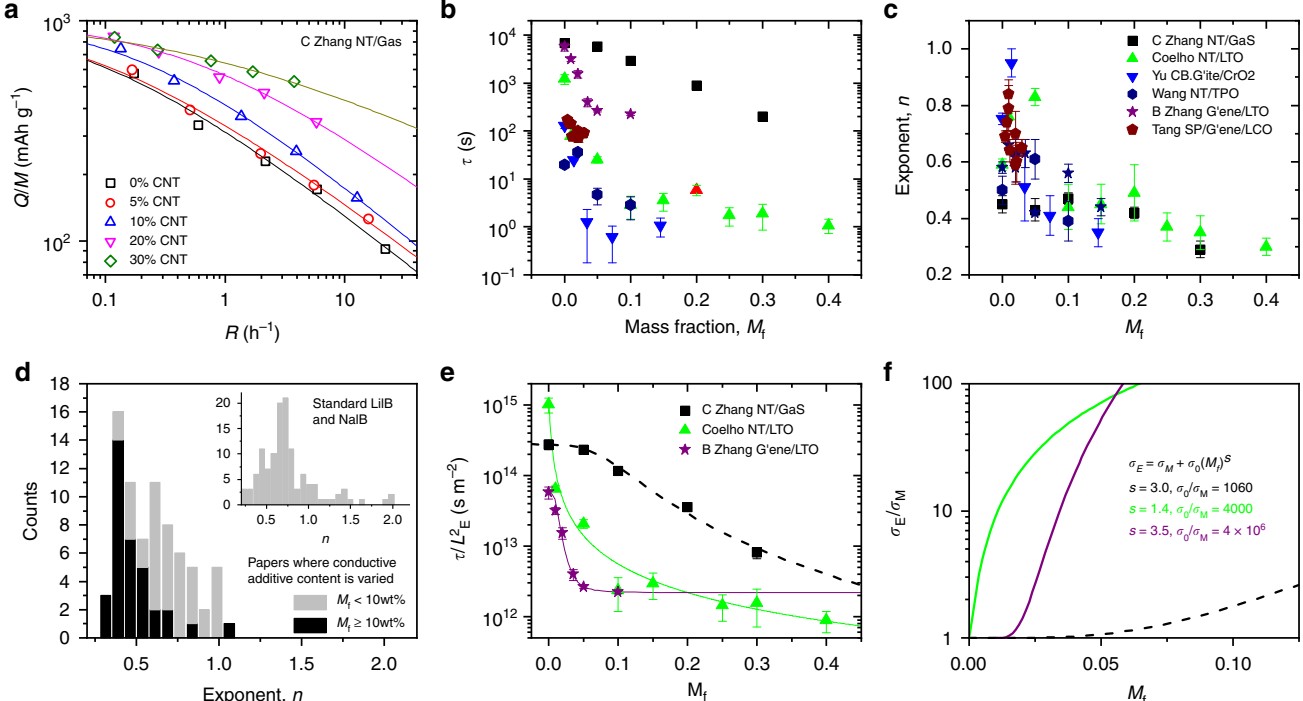

**Fig. 3** The effect of varying the content of conductive additives. **a** Specific capacity versus rate data for lithium ion anodes based on composites of GaS nanosheets and carbon nanotubes with various nanotube mass fractions[7]. The solid lines are fits to Eq. (2). **b** and **c** Characteristic time (**b**) and exponent (**c**), extracted from six papers (refs. [7,18,65,67–69]), plotted versus the mass fraction, $M_f$, of conductive additive. **d** Histogram ($N = 75$) showing frequency of occurrence of $n$ in studies which varied the conductive additive content. The histogram contains data from the papers in **b**, as well as additional refs. [19,70–73] and is divided between electrodes with high and low $M_f$. The inset replots the data from Fig. 2f for comparison. **e** Data for $\tau/L_E^2$ plotted versus $M_f$ for three selected papers[7,18,67]. The solid lines are fits to Eq. (6a) combined with percolation theory (Eq. (7)). **f** Out of plane conductivity, $\sigma_E$, of composite electrodes normalised to the conductivity of the active material alone, $\sigma_M$. This data is extracted from the fits in (**e**) with the legend giving the relevant parameters. N.B. the legend/colour coding in **c** applies to **b**, **c**, **e**, **f**. All errors in this figure are fitting errors combined with measurement uncertainty

thin film of active material, $L_{AM}$ is the film thickness while for a quasi-spherical particle of radius $r$[20], $L_{AM} = r/3$.

For the electrical contribution, we note that every battery electrode has an associated capacitance[76] that limits the rate at which the electrode can be charged/discharged. This effective capacitance, $C_{eff}$, will be dominated by charge storage but may also have contributions due to surface or polarisation effects[76]. Then, we propose $\tau_{Electrical}$ to be the RC time constant associated with the circuit. The total resistance related to the charge/discharge process is the sum of the resistances due to out-of-plane electron transport in the electrode material ($R_{E,E}$), as well as ion transport, both in the electrolyte-filled pores of the electrode ($R_{I,P}$) and in the separator respectively ($R_{I,S}$). Then, the RC contribution to $\tau$ is given by

$$\tau_{Electrical} = C_{eff}(R_{E,E} + R_{I,P} + R_{I,S}) \qquad (5c)$$

The overall characteristic time associated with charge/discharge is then the sum of capacitive, diffusive and kinetic components:

$$\tau = C_{eff}(R_{E,E} + R_{I,P} + R_{I,S}) + \frac{L_E^2}{D_P} + \frac{L_S^2}{D_S} + \frac{L_{AM}^2}{D_{AM}} + t_c \qquad (5d)$$

We note that this approach is consistent with accepted concepts showing current in electrodes to be limited by both capacitive and diffusive effects[77]. The resistances in this equation can be rewritten in terms of the relevant conductivities ($\sigma$) using $R = L/(\sigma A)$, where $L$ and $A$ are the length and area of the region in question. In addition, both ion diffusion coefficients and conductivities in the pores of the electrode and separator can be related to their bulk-liquid values ($D_{BL}$ and $\sigma_{BL}$) and the porosity,

$P$, via the Bruggeman equation[78], ($D_{Porous} = D_{BL}P^{3/2}$ and $\sigma_{Porous} = \sigma_{BL}P^{3/2}$). This yields

$$\tau = L_E^2\left[\frac{C_{V,eff}}{2\sigma_E} + \frac{C_{V,eff}}{2\sigma_{BL}P_E^{3/2}} + \frac{1}{D_{BL}P_E^{3/2}}\right] + L_E\left[\frac{L_S C_{V,eff}}{\sigma_{BL}P_S^{3/2}}\right] + \left[\frac{L_S^2}{D_{BL}P_S^{3/2}} + \frac{L_{AM}^2}{D_{AM}} + t_c\right]$$

| Term | 1 | 2 | 3 | 4 | 5 | 6 | 7 |

$$(6a)$$

where $C_{V,eff}$ is the effective volumetric capacitance of the electrode (F cm$^{-3}$), $\sigma_E$ is the out-of-plane electrical conductivity of the electrode material, $P_E$ and $P_S$ are the porosities of the electrode and separator, respectively. Here $\sigma_{BL}$ is the overall (anion and cation) conductivity of the bulk electrolyte (S m$^{-1}$). More information on the derivation is given in Supplementary Note 1. We note that although in this work, we will use Eq. (6a) to analyse data extracted using Eq. 2, Eq. (6a) could also be applied to characteristic times obtained with any equation[2,30] which can fit capacity-rate data.

This equation has seven terms which we refer to below as terms 1–7 (as labelled). Terms 1, 2 and 4 represent electrical limitations associated with electron transport in the electrode (1), ion transport in both the electrolyte-filled porous interior of the electrode (2) and separator (4). Terms 3, 5 and 6 represent diffusion limitations due to ion motion in the electrolyte-filled porous interior of the electrode (3) and separator (5), as well as solid diffusion within the active material (6). Term 7 is the characteristic time associated with the kinetics of the electro-chemical reaction. We note that, as outlined below, for a given

electrode, not all of these seven terms will be important. We can also write the equation with compound parameters, $a$, $b$ and $c$ to simplify discussion later:

$$\tau = aL_E^2 + bL_E + c \qquad (6b)$$

If Eq. (6a) is correct, then the falloff in $\tau$ with $M_f$ observed in Fig. 3b must be associated with term 1, via the dependence of $\sigma_E$ on $M_f$, which we can express using percolation theory[33]: $\sigma_E \approx \sigma_M + \sigma_0(M_f)^s$, where $\sigma_M$ is the conductivity of the active material, and $\sigma_0$ and $s$ are constants (we approximate the conductivity onset to occur at $M_f = 0$ for simplicity). This allows us to write Eq. (6a) as

$$\tau/L_E^2 \approx \frac{C_{V,eff}/2}{\sigma_M + \sigma_0(M_f)^s} + \beta_1 \qquad (7)$$

where $\beta_1$ represents terms 2–7. We extracted the most extensive data sets from Fig. 3b and reproduced them in Fig. 3e. We find very good fits, supporting the validity of Eqs. (6a) and (7). From the resultant fit parameters (see inset in Fig. 3f), we can work out the ratio of composite to matrix (i.e. active material) conductivities, $\sigma_E/\sigma_M$, which we plot versus $M_f$ in Fig. 3f. This shows that significant conductivity differences can exist between different conductive fillers, leading to different rate performances. As shown in the Supplementary Note 2, by estimating $C_{V,eff}$, we can find approximate values of $\sigma_M$ and $\sigma_0$ which are in line with expectations.

**Thickness dependence.** Equation (6a) implies a polynomial thickness dependence, rather than the $L_E^2$ dependence crudely suggested by Fig. 2c. To test this, we identified a number of papers that reported rate dependence for different electrode thicknesses, as well as preparing some electrodes (see Supplementary Methods) and performing measurements ourselves. An example of such data is given in Fig. 4a for LiFePO$_4$-based lithium ion cathodes of different thicknesses[17], with fits to Eq. (2) shown as solid lines. We fitted eight separate electrode thickness/rate-dependent data sets to Eq. (2) with the resultant $\tau$ and $n$ values plotted in Fig. 4b. Shown in Fig. 4c is $\tau$ plotted versus $L_E$ for each material with a well-defined thickness dependence observed in each case. We fitted each curve to Eq. (6b), finding very good fits for all data sets, and yielding $a$, $b$ and $c$.

We first consider the $c$ parameter (from Eq. (6a), $c = L_S^2/(D_{BL}P_S^{3/2}) + L_{AM}^2/D_{AM} + t_c$). With the exception of µ-Si/NT ($c = 2027 \pm 264$ s) and NMC/NT ($c = 3.6 \pm 1$ s), the fits showed $c \sim 0$ within error. Because the fifth term in Eq. (6a) is always small (typically $L_S \sim 25$ µm, $D_{BL} \sim 3 \times 10^{-10}$ m$^2$ s$^{-1}$ and $P_S \sim 0.4$, yielding ~1 s) and assuming fast reaction kinetics (term 7), $c$ is approximately given by $c \approx L_{AM}^2/D_{AM}$ and so is reflective of the contribution of solid-state diffusion to $\tau$ (term 6). Thus, the high values of $c$ observed for the µ-Si samples are probably due to their large particle size (radius, $r \sim 0.5-1.5$ µm measured by SEM). Combining the value of $c = 2027$ s with reported diffusion coefficients for nano-Si ($D_{AM} \sim 10^{-16}$ m$^2$ s$^{-1}$)[79], and using the equation above with $L_{AM} = r/3$[ref 20], allows us to estimate $r = 3L_{AM} \approx 3\sqrt{cD_{AM}} \sim 1.3$ µm, within the expected range.

That $c \sim 0$ for most of the analysed data can be seen more clearly by plotting $\tau/L_E$ versus $L_E$ in Fig. 4d for a subset of the data (to avoid clutter). These data clearly follow straight lines with non-zero intercepts which is consistent with $c = 0$ and $b \neq 0$ (from Eq. (6a), $b = L_S C_{V,eff}/(\sigma_{BL}P_S^{3/2})$). The second point is important as it can only be the case in the presence of resistance limitations (the $b$ parameter is associated with resistance limitations due to ion transport in the separator).

We extracted the $a$ and $b$ parameters from the fits in Fig. 4c and plotted $a$ versus $b$ in Fig. 4e. The significance of this graph can be seen by noting that we can combine the definitions of $a$ and $b$ in Eq. (6b) to eliminate $C_{V,eff}$, yielding

$$a = \left[\frac{\sigma_{BL}P_S^{3/2}}{\sigma_E} + \left(\frac{P_S}{P_E}\right)^{3/2}\right]\frac{b}{2L_S} + \frac{1}{D_{BL}P_E^{3/2}} \qquad (8)$$

The value of $D_{BL}$ tends to fall in a narrow range $(1-5) \times 10^{-10}$ m$^2$ s$^{-1}$ for common battery electrolytes[80,81]. Taking $D_{BL} = 3 \times 10^{-10}$ m$^2$ s$^{-1}$ and using $L_S = 25$ µm (from the standard Celgard separator)[82], we plot Eq. (8) on Fig. 4e for two scenarios with extreme values of separator[83]/electrode porosity and different bulk-electrolyte to electrode conductivity ratios (see panel). We find the data to roughly lie between these bounds. This shows the effect of electrode and separator porosities and identifies the typical range of $\sigma_{BL}/\sigma_E$ values. In addition, because electrolytes tend to have $\sigma_{BL} \sim 0.5$ S m$^{-1}$[19], this data implies the out-of-plane electrode conductivities to lie between 0.2 and 10 S m$^{-1}$ for these samples. To test this, we measured the out-of-plane conductivity for one of our electrodes ((NMC/~1%NT)), obtaining 0.3 S m$^{-1}$, in good agreement with the model. Interestingly, the $a$ value for the GaS/NT electrodes of Zhang et al.[7] is quite large, suggesting a low out-of-plane conductivity. This is consistent with the NT $M_f$ dependence (Fig. 3f), taken from the same paper, which indicates relatively low conductivity enhancement in this system.

From the definition of $b$ (Eq. (6a), $b = L_S C_{V,eff}/(\sigma_{BL}P_S^{3/2})$), we can estimate the effective volumetric capacitance, $C_{V,eff}$, for each material (estimating $\sigma_{BL}$ from the paper and assuming $P_S = 0.4$[83] and $L_S = 25$ µm unless stated otherwise in the paper). Values of $C_{V,eff}$ vary in the range $\sim 10^3 - 10^5$ F cm$^{-3}$. To put this in context, typical commercial batteries have capacitances of ~1500 F (18,650 cylindrical cell)[84]. Assuming the electrodes act like series capacitors, gives a single-electrode capacitance of ~3000 F. Approximating the single-electrode volume as ~25% of the total yields an electrode volumetric capacitance of ~10$^3$ F cm$^{-3}$, similar to the lower end of our range.

We found these $C_{V,eff}$ values to scale linearly with the intrinsic volumetric capacity of each material ($Q_V = \rho_E Q_M$, where $\rho_E$ is the electrode density) as shown in Fig. 4f, indicating the capacitance to be dominated by charge storage effects. This relationship can be written as $C_{V,eff}/Q_V = 1/V_{eff}$, where $V_{eff}$ is a constant. Fitting shows $C_{V,eff}/Q_V = 1/V_{eff} = 28$ F/mAh, a relationship which will prove useful for applying the model.

**Other tests of the characteristic time equation.** We can also test the veracity of Eq. (6a), in other ways. The data of Yu et al.[16] for electrodes with different conductivities, which was shown in Fig. 4c, has been replotted in Fig. 5a as $\tau/L_E$ versus $L_E$ and shows these composites to have roughly the same value of $b$ (intercept) but significantly different values of $a$ (slope). This is consistent with the electrode conductivity effecting term 1 in Eq. (6a), perfectly in line with the model.

We can also test the porosity dependence predicted by Eq. (6a), although electrodes with varying porosity also tend to display varying conductivity, making it difficult to isolate the porosity dependence. However Bauer et al.[19] describe rate performance of graphite/NMC electrodes with different porosities yet the same conductivity. Shown in Fig. 5b are $\tau/L_E^2$-values, found by fitting their data, plotted versus porosity. Eq. (6a) predicts that this data should follow

$$\frac{\tau}{L_E^2} = \left[\frac{C_{V,eff}}{2\sigma_{BL}} + \frac{1}{D_{BL}}\right]P_E^{-3/2} + \beta_2 \qquad (9)$$

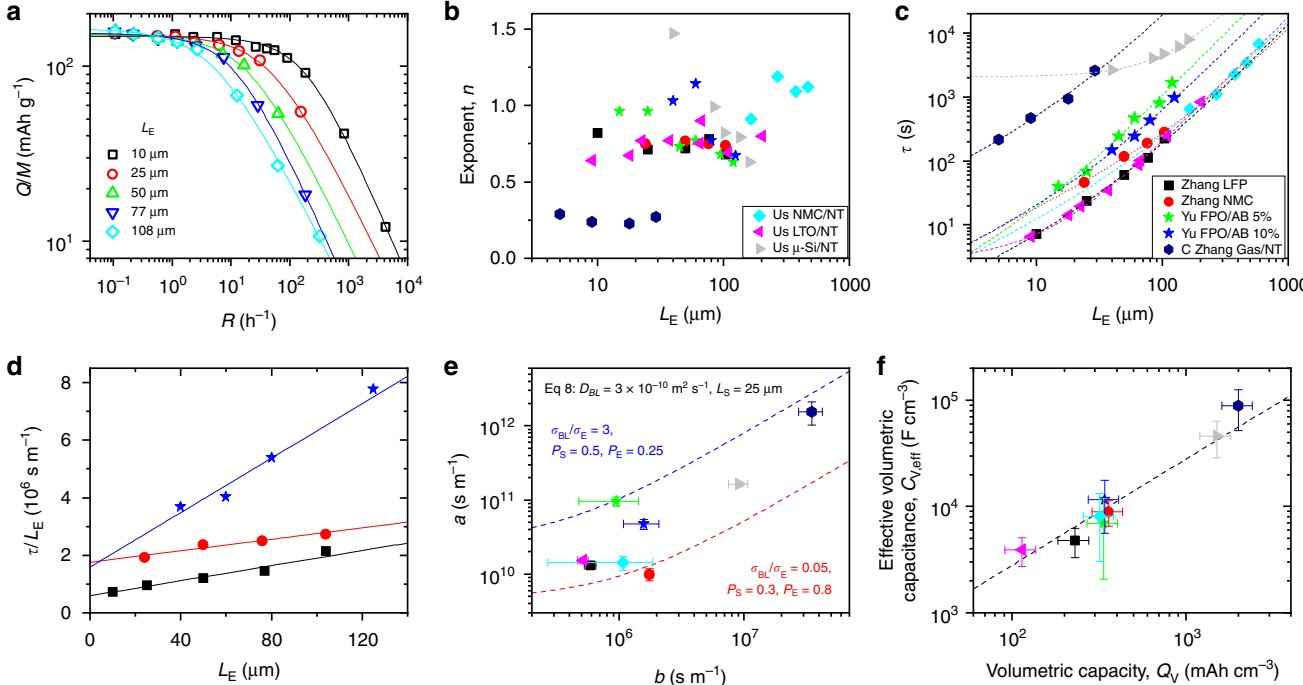

**Fig. 4** The effect of varying electrode thickness. **a** Specific capacity versus rate data for LiFePO$_4$-based lithium ion cathodes of different thicknesses[17]. The solid lines are fits to Eq. (2). **b** and **c** Exponent (**b**) and characteristic time (**c**) plotted versus electrode thickness for eight data sets including three measured by us and five from the literature[7,16,17]. The legends in **b** and **c** both apply to panels **b**–**f**. The dashed lines in **c** are fits to the polynomial given in Eq. (6b). **d** Plots of $\tau/L_E$ versus $L_E$ for a subset of the curves in (**c**), showing the c-terms to be negligible (true for all data in (**c**) except the μ-Si/NT and NMC/NT data sets). **e** a parameter plotted versus b parameter (see Eq. (6a), (6b)) for the data in **c**. The lines are plots of Eq. (8) using the parameters given in the panel and represent limiting cases. **f** Effective volumetric capacitance, estimated from the b parameters, plotted versus the volumetric capacity, $Q_V = \rho_E Q_M$. The dashed line is an empirical curve which allows $C_{V,eff}$ (F cm$^{-3}$) to be estimated from $Q_V$ (mAh cm$^{-3}$):$C_{V,eff}/Q_V = 28$ F/mAh. All errors in this figure are fitting errors combined with measurement uncertainty

where $\beta_2$ represents terms 1 and 4−7. Combining the fit parameters with estimates of $C_{V,eff}$ and $D_{BL}$ (see Supplementary Note 3) yields a value of $\sigma_{BL} = 0.5$ S m$^{-1}$, in line with typical values of ~0.1−1 S m$^{-1}$[9].

Yu et al.[16] reported rate dependence for LiFePO$_4$ electrodes with various electrolyte concentrations, $c$. Shown in Fig. 5c are $\tau/L_E^2$-values, found by fitting their data, plotted versus $1/c$. We can model this crudely by replacing the electrolyte conductivity, $\sigma_{BL}$, in Eq. (6a), using the Nearnst–Einstein equation, $\sigma_{BL} \approx F^2 c D_{BL}/t^+ R_G T$ as a rough approximation (here $t^+$ is the cation transport number which allows conversion between overall conductivity, $\sigma_{BL}$, and cation diffusion coefficient, $D_{BL}$, $R_G$ is the gas constant and the other parameters have their usual meaning). Then Eq. (6a), predicts

$$\frac{\tau}{L_E^2} = \frac{t^+ R_G T}{F^2 D_{BL} c}\left[\frac{C_{V,eff}}{2P_E^{3/2}} + \frac{L_S}{L_E}\frac{C_{V,eff}}{P_S^{3/2}}\right] + \beta_3 \tag{10}$$

where $\beta_3$ represents terms 1, 3 and 5−7. Fitting the data and estimating the various parameters as described in Supplementary Note 3 allows us to extract $D_{BL} \approx 6 \times 10^{-11}$ m$^2$ s$^{-1}$, close to the expected value of ~$10^{-10}$ m$^2$ s$^{-1}$.

In addition, we varied the separator thickness ($L_S$) by using one, two and three stacked separators, measuring the rate performance of NMC/0.5%NT electrodes in each case. Values of $\tau/L_E^2$ extracted from the fits are plotted versus $L_S$ in Fig. 5d. Then Eq. (6a), (6b) predicts

$$\frac{\tau}{L_E^2} = L_S\left[\frac{C_{V,eff}}{L_E \sigma_{BL} P_S^{3/2}}\right] + \beta_4 \tag{11}$$

where $\beta_4$ represents terms 1−3 and 5−7. Fitting the data and estimating parameters (see Supplementary Note 3) yields $\sigma_{BL} \sim 0.6$ S m$^{-1}$, very similar to typical values of ~0.1−1 S m$^{-1}$[9].

Eq. (6a) would imply the solid-state diffusion term (term 6) could be significant if $D_{AM}$ were small, especially for low-$L_E$ electrodes. Ye et al.[12] measured rate dependence of electrodes consisting of thin nano-layers (<20 nm) of anatase TiO$_2$ deposited on highly porous gold current collectors. In these systems, we expect solid-state diffusion to be limiting. The $\tau$ values found by fitting their data are plotted versus the TiO$_2$ thickness in Fig. 5e. Examining Eq. (6a), (6b), we would expect this data to be described by

$$\tau = \frac{L_{AM}^2}{D_{AM}} + \beta_5 \tag{12}$$

where $\beta_5$ represents terms 1−5 and 7. This equation fits the data very well, yielding a solid-state diffusion coefficient of $D_{AM} = 3.3 \times 10^{-19}$ m$^2$ s$^{-1}$, close to values of $(2-6) \times 10^{-19}$ m$^2$ s$^{-1}$ reported by Lindstrom et al.[85].

## Discussion

This work shows that Eqs. (2) and (6a) fully describe capacity-rate data in battery electrodes. This model can be applied in a number of ways, with the simplest being to fit experimental data to find $\tau$ and then use Eq. (6a) to analyse the dependence of $\tau$ on other variables. We note the ability to fit data is a major advantage over more sophisticated models.

We can also use Eq. (6a) to understand the balance of the different contributions to rate performance and so to design better electrodes. Earlier, we introduced the transport coefficient,

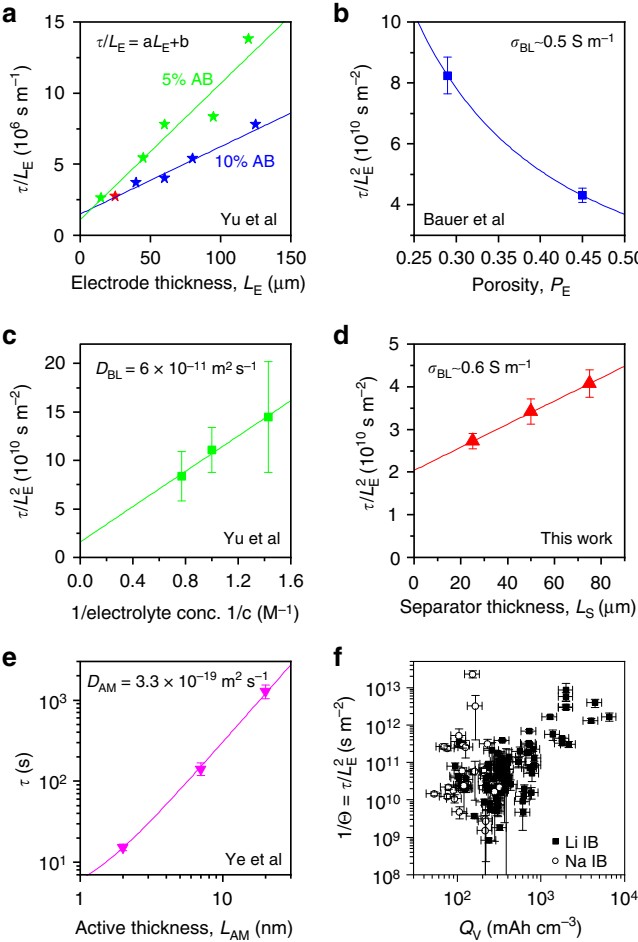

**Fig. 5** Further testing of the terms in Eq. (6a), (6b). **a** $\tau/L_E$ versus $L_E$ for electrodes with 5% and 10% acetylene black, and so different conductivities (extracted from ref. [16]). This results in different $a$ parameters (slopes) but the same $b$ parameter (intercept), consistent with Eq. (6a), (6b). **b** $\tau/L_E^2$ versus porosity extracted from ref. [19]. The line is a fit to Eq. (9) and yields a value of $\sigma_{BL}$ close to the expected value (see panel). **c** $\tau/L_E^2$ versus inverse electrolyte concentration extracted from ref. [16]. The line is a fit to Eq. (10) and yields $D_{BL}$ close to the expected value (see panel). **d** $\tau/L_E^2$ versus separator thickness (this work). The line is a fit to Eq. (11) and yields $\sigma_{BL}$ close to the expected value (see panel). **e** Characteristic time versus the thickness of a thin active layer ($TiO_2$) extracted from ref. [12]. The line is a fit to Eq. (12) and yields a diffusion coefficient for Li ions in anatase $TiO_2$ close to the expected value[85]. **f** $1/\Theta$ plotted versus the intrinsic volumetric electrode capacity, $Q_V$, for cohorts I and II showing the scaling predicted by Eq. (13a). All errors in this figure are fitting errors combined with measurement uncertainty

$\Theta$, as a metric for rate performance. Applying Eq. (6a), we find:

$$1/\Theta = \frac{\tau}{L_E^2} = \frac{C_{V,eff}}{2\sigma_E} + \frac{C_{V,eff}}{2\sigma_{BL}P_E^{3/2}} + \frac{1}{D_{BL}P_E^{3/2}} + \frac{C_{V,eff}L_S/L_E}{\sigma_{BL}P_S^{3/2}} + \frac{L_S^2/L_E^2}{D_{BL}P_S^{3/2}} + \frac{L_{AM}^2/L_E^2}{D_{AM}} + \frac{t_c}{L_E^2}$$

| Term | 1 | 2 | 3 | 4 | 5 | 6 | 7 |
|------|---|---|---|---|---|---|---|

(13a)

Because $L_E$ has either been eliminated or mostly appears as a ratio with other lengths, $\Theta$ is semi-intrinsic to the electrode/electrolyte system and the natural descriptor of rate performance, incorporating diffusive, electrical and kinetic limitations. Because rate performance is maximised when $\tau$ is small we can consider

$\Theta$ as a figure of merit for electrodes, with larger values of $\Theta$ indicating better rate performance. Thus, any strategy to improve rate performance must focus on maximising $\Theta$. Values of $\Theta$ can be put in context by Fig. 2d which show the practical upper limits to be $\Theta \sim 10^{-9}$ m$^2$ s$^{-1}$.

Writing Eq. (13a) in this way allows another test of our model as it predicts $1/\Theta$ to scale with $C_{V,eff}$. Because $C_{V,eff} \propto Q_V$, we can test this prediction by plotting $1/\Theta$ versus $Q_V$ in Fig. 5f. We find a well-defined (and as far we know completely unknown) relationship, adding further support to our model. This graph is important as it confirms the influence of $C_{V,eff}$ on electrode rate performance while highlighting the unfortunate fact that high-performance electrode materials appear to have an inherent disadvantage in terms of rate behaviour. In addition, sodium and lithium battery data overlap, suggesting sodium electrodes to be predominantly limited by capacitive effects rather than solid state diffusion as is usually believed[74].

It is important to realise what parameters are controllable during optimisation. $D_{BL}$ is limited by solvent effects, while $\sigma_{BL}$ is typically maximised at $\sim 0.5$ S m$^{-1}$.[9] $D_{AM}$ and $C_{V,eff}$ (via $Q_V$) are set by materials choice. $L_S$ and $P_S$ are controllable but limited by separator availability. While $L_E$ is controllable, enhancement of capacity will require its maximisation. This means $\sigma_E$, $P_E$ and $L_{AM}$ are the only truly free parameters for optimisation.

Eq. (13a) also gives insight into parameter optimisation. All seven terms must be minimised for battery electrodes to display maximised rate performance (i.e. maximal $\Theta$). In Fig. 6, we have used Eq. (13a) to plot the values of $1/\Theta$ for each term, as well as their sum versus five electrode parameters, $L_E$, $C_{V,eff}$, $\sigma_E$, $L_{AM}$ and $P_E$, using typical values for the remaining parameters (see panel). To avoid confusion, we plot $\tau$ (rather than $1/\Theta$) versus $L_E$ in Fig. 6a. This shows solid diffusion to dominate thin electrodes (term 6) but electrical limitations associated with ions in electrode pores to be dominant for electrodes thicker than $\sim 50$ μm (term 2). In panels b–e, we plot $1/\Theta$ as a function of each parameter. We find electrical limitations to be important for high-capacity electrode materials which also display high $C_{V,eff}$ (Fig. 6b). As shown in Fig. 6c, it is important to maximise the (out-of-plane) conductivity to minimise its contribution to $1/\Theta$. In thick electrodes, the effect of solid diffusion (Fig. 6d) is only important for the largest active-material particles. Interestingly, changing the electrode porosity (Fig. 6e) has a relatively small impact on $\Theta$. In addition, we note that term 5 is always small and can generally be neglected. In addition, taking a relatively high[20] value of $t_c = 25$ s, gives a reaction kinetics contribution (term 7) which is negligible compared to other terms (although reaction kinetics can be rate limiting for thin electrodes[86]).

Eq. (13a) can be simplified considerably for electrodes with thickness >100 μm, as found in practical cells. Then, $\Theta$ is dominated by terms 2 and 4 with a non-negligible contribution from term 3 under certain circumstances. Specifically, because terms 1 and 2 scale in similar ways, term 1 can be ignored when it is much smaller than term 2, i.e. if $\sigma_E \gg \sigma_{BL}P_E^{3/2}$. Taking $\sigma_{BL} \sim 0.5$ S m$^{-1}$ and $P_E \sim 0.5$, this is true if $\sigma_E \gg 1$ S m$^{-1}$, which should be the aim when introducing conductive additives. Term 6 can be neglected so long as it is much smaller than the ubiquitous term 3, i.e. if $L_E/L_{AM} \gg \sqrt{D_{BL}P_E^{3/2}/D_{AM}}$. For $r = 60$ nm ($L_{AM} = 20$ nm) Si particles ($D_{AM} \sim 10^{-16}$ m$^2$ s$^{-1}$)[79], this is true if $L_E \gg 20$ μm which will apply in commercial electrodes. In addition, we neglect term 7 as $t_c/L_E^2$ should become relatively small for thick electrodes.

Under these circumstances, terms 1, 5, 6 and 7 in Eq. (13a) are negligible, giving an approximate expression for $\Theta$. This equation can be generalised and simplified further by using the Nearnst–Einstein equation to eliminate $\sigma_{BL}$, allowing us to

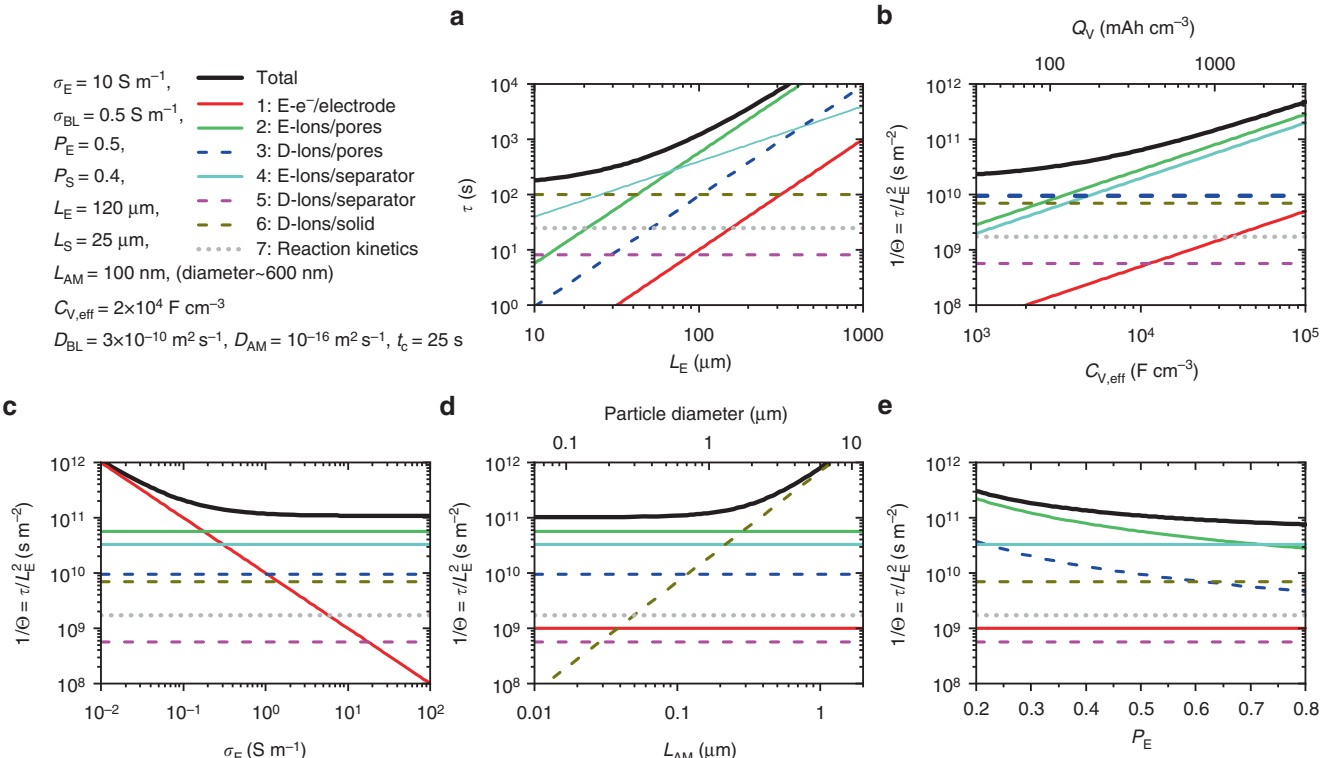

**Fig. 6** Comparison of the magnitude of terms 1−7 in Eq. (13a) as well as their sum, for a range of electrode parameters. Note that, while in **a**, $\tau$ is plotted versus $L_E$, in all other panels, $1/\Theta$ is plotted versus the relevant parameter. The parameters used are given at the top left. Those bold parameters were kept constant in all panels except one, where they were varied. The solid black lines represent the total value of $\tau$ or $1/\Theta$. Low values of both $\tau$ and $1/\Theta$ are needed for good rate performance. The other curves represent the seven individual terms in Eq. (13a), labelled 1−7 (numbered from left to right in the equation). Electrical and diffusion limited terms are marked as solid and dashed lines, respectively, with the reaction kinetics term represented by grey dots. The legend in the top left gives the term number as well of a summary of what it represents. Those terms labelled by "E" are electrically limited while those labelled by "D" are diffusion limited. The top axis in **b** represents the volumetric capacity of the electrode calculated using $C_{V,eff}/Q_V = 28$ F/mAh. N.B., $L_{AM} = 100$ nm corresponds to a particle diameter of $2r \approx 600$ nm because $L_{AM} = r/3$ for pseudo-spherical particles

express $\Theta$ in terms of the electrolyte concentration, $c$, for thick electrodes:

$$\Theta \approx \frac{D_{BL}P_E^{3/2}}{1 + \frac{t^+ R_G T C_{V,eff}}{2F^2 c}\left(1 + 2\frac{L_S}{L_E}\left(\frac{P_E}{P_S}\right)^{3/2}\right)} \qquad (13b)$$

Inspection of Eq. (13b) shows the maximum possible value of $\Theta$ is achieved when $C_{V,eff}$ is small and the electrode is limited solely by diffusion of ions in the electrolyte-filled pores of the electrode: $\Theta_{max} \approx D_{BL}P_E^{3/2}$, which could reach ~3 × $10^{-10}$ m² s⁻¹ in high porosity electrodes. This value of $\Theta_{max}$ represents the basic rate limit for the electrode and is indicated in Fig. 2d by the arrow. Virtually all of the electrodes analysed in this work show $\Theta < \Theta_{max}$. Interestingly, a recent paper, which fabricated nanostructured electrodes with the aim of achieving ultrafast charge/discharge[21], reported data consistent with $\Theta \sim 3 \times 10^{-10}$ m² s⁻¹, very close to the maximum value suggested by our work.

In conclusion, we have developed a quantitative model to describe rate performance in battery electrodes. This combines a semi-empirical model for capacity as a function of rate with simple expressions for the diffusive, electrical and kinetic contributions to the characteristic time associated with charge/discharge. This model is completely consistent with a wide range of results from the literature and allows quantitative analysis of data by fitting to yield numerical values of parameters, such as electrode conductivity and diffusion coefficients.

## Methods

The capacity versus rate data from the literature sources were extracted using the "Digitizer" function in Origin. All fitting was performed using Origin software (here we used Origin version 2015–2018) via the "Nonlinear Curve Fit" function, according to the model equation. Care must be taken in fitting, with more detailed information given in Supplementary Methods and Supplementary Notes 2 and 3. All fits are shown in Supplementary Figures while all data is listed in supplementary data files.

We also prepared several electrode sets. Those electrodes were prepared by a conventional slurry casting method. The electrochemical properties of the electrodes were measured in 2032-type coin cells (MTI Corp.) with a half-cell configuration using Li-metal as a counter electrode. More details are described in Supplementary Methods.

## Data availability

The source data underlying Figs. 2–6, Supplementary Figs. 1–41 and Supplementary Tables 1–6 are provided as a Source Data file.

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

## Acknowledgements

All authors acknowledge the SFI-funded AMBER research centre (SFI/12/RC/2278) and Nokia for support. J.N.C. thanks Science Foundation Ireland (SFI, 11/PI/1087) and the Graphene Flagship (grant agreement no. 785219) for funding. V.N. thanks the European Research Council (SoG 3D2D Print) and Science Foundation Ireland (PIYRA) for funding. Dr. Ruiyuan Tian thanks Dr. Chuanfang (John) Zhang and Dr. Sebastian Barwich for useful discussions.

## Author contributions

R.T. collected and catalogued the literature data. R.T, P.J.K., G.C. and J.C. performed experiments. R.T. S-H.P., V.N. and J.N.C analysed the data and developed the model. J.N.C. conceived the project and wrote the paper with help from R.T. and S.-H.P. All authors discussed the results and commented on the manuscript.

## Additional information

**Competing interests:** The authors declare no competing interests.

