## [Peer Review File · Nature Communications]

Reviewers' comments:

Reviewer #1 (Remarks to the Author):

The work of Tian et al. is based on quantifying various parameters associated with rate-performance of battery electrodes. This work is extensively detailed with careful assumptions made in terms of identifying key limiting factors of rate performance. I recommend this publication for acceptance with minor comments, appended below.

Page 3: does "intrinsic" mean "theoretical"? Based on the equation 2, illustrated in Fig. 1a, the intrinsic capacity seems to be the capacity at slow rate. Although this is graphically shown in Fig. 1c, this should be stated more explicitly in writing because the first assumption I made when I saw "intrinsic" was that this might be the "theoretical" capacity.

Page 3: Define M in Equation 1. Even though by "specific" most people understand this to be mass-normalized, it would be beneficial to just explicitly mention this. And also, is this mass of the electrode, mass of active mass, or mass of total device? Can it apply to all three based on the system that is being studied?

Page 6: During the explanation of $n > 1$, the author mentioned unwanted reactions such as alloying or Li-plating. I would like to suggest that there may be other side reactions that affect n such as continuous SEI formation caused by particle fracture upon repeated cycling. As a follow-up question, what is the physical interpretation of $n < 0.5$? Would this indicate bulk mass transport limitations? The number of points below 0.5 is not insignificant so this should be addressed.

Page 16: Although LFP-based cathodes and Si-based cathodes are different systems, the electrode thicknesses are being compared and for the case of LFP, regardless of the electrode thickness, the low-rate capacity is relatively constant whereas for Si, the capacity is higher. Is this because of the volume expansion of Si is much higher so at slow rates or could this be a result of greater SEI formation at those slow rates.

Reviewer #2 (Remarks to the Author):

I do not recommend this article for publication. The authors propose a semi-empirical model to represent capacity offset in rechargeable batteries. Although there is some merit in having an analytic expression for capacity with rate as opposed to a detailed physical model, it is not an appropriate topic for Nature Communications.

There are some misstatements about existing batteries models, most lithium ion model derive from the Doyle, Fuller, Newman approach. The abstract states "...no quantitative model exists which can be used to fit data to give insights into the dominant rate-limiting processes in a given electrode-electrolyte system" This is simply not true. On line 60, referring to existing models "all are limited in that they only describe a single rate limiting mechanism." Existing physics based models include numerous phenomena and characteristic times.

Reviewer #3 (Remarks to the Author):

This manuscript presents very useful modeling strategies for predicting the rate characteristics of batteries. Although is prior literature on this particular topic, the present manuscript presents a sufficiently novel and useful approach to warrant publication. Otherwise, the paper is well written, thorough, and the methods and results well documented. I find no issues that need to be addressed

prior to publication in "Nature Communications". This will be an impactful paper for the scientific community that is working on energy-storage technologies.

Reviewer #1

The work of Tian et al. is based on quantifying various parameters associated with rate-performance of battery electrodes. This work is extensively detailed with careful assumptions made in terms of identifying key limiting factors of rate performance. I recommend this publication for acceptance with minor comments, appended below.

Page 3: does “intrinsic” mean “theoretical”? Based on the equation 2, illustrated in Fig. 1a, the intrinsic capacity seems to be the capacity at slow rate. Although this is graphically shown in Fig. 1c, this should be stated more explicitly in writing because the first assumption I made when I saw “intrinsic” was that this might be the “theoretical” capacity.

Yes, the reviewer is correct, this is the capacity at low rate. We have clarified this in relation to both equations 1 and 2.

Page 3: Define M in Equation 1. Even though by “specific” most people understand this to be mass-normalized, it would be beneficial to just explicitly mention this. And also, is this mass of the electrode, mass of active mass, or mass of total device? Can it apply to all three based on the system that is being studied?

This is a good point which we should have clarified. This equation can apply to any type of capacity (eg C/M_{total} , C/M_{active} , C/A , C/V etc) so long as the C/X and C_X parameters match. Ie if the parameter on the LHS of eq 2 is C/A , the first parameter on the RHS should be C_A . We have modified the text to clarify:

“Here C/M is the measured, rate-dependent specific capacity (i.e. normalised to electrode mass), C_M is the specific capacity at low rate and τ is the characteristic time associated with charge/discharge. We note that, although we have written equation 2 in terms of specific capacity, it could also represent areal capacity, volumetric capacity etc, so long as C/M is replaced by the relevant measured parameter (e.g measured areal capacity, C/A) while C_M is replaced by the low-rate value of the that parameter (which we might denote C_A).”

Page 6: During the explanation of $n>1$, the author mentioned unwanted reactions such as alloying or Li-plating. I would like to suggest that there may be other side reactions that affect n such as

continuous SEI formation caused by particle fracture upon repeated cycling. As a follow-up question, what is the physical interpretation of $n < 0.5$? Would this indicate bulk mass transport limitations? The number of points below 0.5 is not insignificant so this should be addressed.

Answer: We agree, here the term “alloying effect” also includes exactly what the reviewer means. Materials with alloying mechanism (i.e. Si or Sn) that have large volume change (up to 300%), might lead the pulverization of active materials with continuous, unstable SEI formation upon repeated cycling. Thus all these phenomena negatively affect the rate-capability, resulting $n > 1$, which means a rate-limiting mechanism is even more severe than electrical limitations.

We have modified the text to reflect this: “We note that the highest values of n are associated with Si-based electrodes where unwanted electrochemical effect such as alloying, Li-plating or continuous SEI formation, caused by particle pulverisation, may affect lithium storage kinetics.⁶⁹”

It is not clear why n should be < 0.5 . Mass transport limitation should yield $n = 0.5$. I agree that we should give an explanation but to be honest, we don't have a good one. However, having checked the data, we note that some of the data sets which showed $n < 0.5$ were quite limited, showing small capacity falloff at high rate. We have modified the text to read: “In addition, it is unclear why some data points are consistent with $n < 0.5$, although this may represent a fitting error associated with datasets showing small capacity falloffs at higher rate.”

Page 16: Although LFP-based cathodes and Si-based cathodes are different systems, the electrode thicknesses are being compared and for the case of LFP, regardless of the electrode thickness, the low-rate capacity is relatively constant whereas for Si, the capacity is higher. Is this because of the volume expansion of Si is much higher so at slow rates or could this be a result of greater SEI formation at those slow rates.

I don't really understand the question. The low-rate capacity of LFP is lower than that of Si, simply because the theoretical capacity of Si is much higher (3500 mAh/g versus 170 mAh/g).

Reviewer #2

I do not recommend this article for publication. The authors propose a semi-empirical model to represent capacity offset in rechargeable batteries. Although there is some merit in having an analytic expression for capacity with rate as opposed to a detailed physical model, it is not an appropriate topic for Nature Communications.

The reviewer says “*there is some merit in having an analytic expression for capacity with rate as opposed to a detailed physical model,*” We would like to point out that this work does not just consist of an analytic expression for capacity with rate (i.e. equation 2). Much more important is equation 6a which is a simple physical model for the characteristic time (which is extracted from the capacity – rate data with equation 2). Equation 6a relates the characteristic time (τ) to the main properties of the battery system under study eg electrode conductivity, electrolyte diffusivity, active particle size etc. Importantly, our model is consistent with all experimental data we have tested it against (eg figs 3,4,5).

This model allows capacity-rate data, which is the output of the most common rate test in batteries, to be fully linked to physical properties. Most importantly, both equations 2 and 6a are simple analytical functions which can be fitted to data using standard graphing packages.

For example, a number of papers present C v rate data for different electrode thicknesses. Our approach allows the C v rate data to be fitted to get the characteristic time. This parameter can then be plotted versus electrode thickness and the data fitted using equation 6a. The procedure outlined above applies not just to electrode thickness but other parameters such as electrode conductivity, porosity, particle size and separator thickness. Fitting τ versus any of these parameters allows data to be extracted such as electrolyte conductivity and diffusion coefficient.

We fully accept that the model presented here is simplistic compared to more sophisticated models such as that pioneered by Doyle/Fuller/Newman (DFN). However, it is a good basic tool for first order analysis or indeed for benchmarking of performance. We agree that “*a detailed physical model*” such as DFN’s provides a more advanced physical understanding of battery performance. However, there are also advantages associated with simplicity, especially if it allows the average researcher to fit their data using a standard graphing package.

We respectively disagree with the opinion of the reviewer that this work “*is not an appropriate topic for Nature Communications.*”

At present, sophisticated physical models for rate performance are not generally used by experimentalists to analyse their data, probably because they are not user-friendly. We very briefly checked 49 experimental papers reporting rate data on battery electrodes (essentially a subset of the papers cited in this manuscript). Only one (J Electrochem Soc 163, A138-A149 (2016)), used sophisticated models such as DFN’s to analyse their data. This is not a negative reflection on these models, just an indication that most experimentalists probably don’t feel able to implement them.

Our vision is that access to a simple analytical model which can fit data using standard graphing packages would allow all researchers to perform quantitative analysis on their rate data, whether that

be for benchmarking, physical analysis or predictive work. We believe this would have a significant impact on the battery field as a whole and as a result, we are convinced that this is an appropriate topic for Nature Communications.

There are some misstatements about existing batteries models, most lithium ion model derive from the Doyle, Fuller, Newman approach. The abstract states "...no quantitative model exists which can be used to fit data to give insights into the dominant rate-limiting processes in a given electrode-electrolyte system" This is simply not true.

We are of course aware of physical models such as those based on the Doyle, Fuller, Newman approach. To me the operative word is "fit". I assume the reviewer is saying here that such models can be used to fit experimental data. For me, the problem here is one of terminology. When I used the term "to fit data" I meant using a standard software package such as origin to optimise fit parameters using something like chi-squared to make a simple analytical function match experimental data.

Conversely, using detailed physical models such as that of Doyle/Fuller/Newman involves solving a set of ~8 coupled equations to generate solutions which match experimental data. This cannot be done in standard packages and is a method which is not accessible to most experimentalists (see comment above). I would generally consider such a procedure as "simulation" and the outputs as simulations rather than fits. Indeed, in some of Newman's early papers (eg J. Electrochem. Soc. 143, 6, 1890) the outputs of such a procedure are referred to as *simulations*. Indeed Doyle and Newman themselves acknowledge that these methods are complicated: "*However, due primarily to their generality, these models are complicated*" (J. Electrochem. Soc. 27 (1997) 846±856).

Now, it is true that Doyle and Newman simplified their differential equations to give three approximate analytical expressions for $C v$ rate which apply for three defined rate-limiting mechanisms. However, there are two problems here. Firstly, when working with experimental materials it is usually not clear what the rate limiting mechanism is. This makes it unclear what equation to use. Secondly, these solutions show well-defined rate dependence (eg $1/I$ at high rate). However, this does not agree with experimental data which tends to show $(C \sim \text{rate}^n$ at high rate with $0.5 < n < 2$). In addition, there are semi-empirical models which incorporate diffusion but no other rate limiting mechanisms.

So, we agree, our phraseology was poor. However, we maintain that a user-friendly, general, fittable model is needed. We have modified the text in the abstract to try to clarify:

“However, no simple, yet comprehensive, model exists which can be used to quantitatively fit capacity-rate data to give insights into the dominant rate-limiting processes in a given electrode-electrolyte system.”

In addition, we have added text (see below) to the introduction to explain more what models exist and what are their limitations.

On line 60, referring to existing models "all are limited in that they only describe a single rate limiting mechanism." Existing physics based models include numerous phenomena and characteristic times.

The problem here is similar to the one above. I was referring to models which can be used to fit data using standard graphing packages. Including the start of the sentence: sentence: “*While a small number of models exist which can be used to fit capacity versus rate data, all are limited in that they only describe a single rate limiting mechanism (e.g. diffusion²⁵ in the electrolyte^{24,26} or solid particles²)*” The cited papers all give semi-empirical equations relating C to rate via single characteristic times associated with diffusion. However, we accept that we could have phrased this better and should have given a broader description of what models are in use. To address this, we have added the following text to the introduction:

“However, during analysis of experimental data, it can be very difficult to quantitatively link the observed rate performance to the factors given above. The most commonly reported experimental rate-performance data are capacity vs. rate curves. Ideally, the experimentalist would be able to fit his/her capacity-rate data to an analytic model which quantitatively includes the influence of the parameters above (i.e. electrode thickness, porosity, particle size etc). However, to the best of our knowledge, comprehensive, fittable, analytic models are not available.

A number of theoretical models which describe Li-ion batteries have been reported.²³ Probably most relevant are the electrochemical models,^{20,24-26} many of which are based on the Doyle- Fuller-Newman (DFN) approach which uses concentrated solution theory to model the charge/discharge process in Li-ion cells.^{27,28} Such models provide a general and comprehensive description of both electrode and cell operation and tend to match well to experimental data.¹⁴ However, these models involve the numerical solution of a number of coupled differential equations and require knowledge of a large number of numerical parameters which may not be available when dealing with new materials. As such, their application is closer to simulation than fitting which makes them relatively complex to use and so inaccessible to the majority of experimentalists. The DFN model has been

simplified by uncoupling the differential equations, which can be achieved under certain limited circumstances. This results in three simple, fittable, analytical models which describe capacity as a function of rate for three different rate-limiting processes: Ohmic limitations and diffusion in electrolyte or active particles.²² However, in experimental systems, the rate-limiting process may not be known, raising questions as to which equation to use. In addition, the functional form of the rate dependence in these simplified solutions is unlikely to match experimental data over the entire range of experimental rates. As a result, these equations are not widely used for fitting purposes. Alternatively, a number of fittable, analytical physical models have been proposed, which are limited in that they only describe the high-rate region.^{24,29} Because of such problems associated with physics-based models, a number of researchers have proposed empirical equations which can be used to fit capacity versus rate data over the whole experimental range. However, all are limited in that they only describe a single rate limiting mechanism, generally diffusion.^{2,30,31}

What is needed is a simple model which experimentalists can use to fit capacity vs. rate data and which quantitatively incorporates all main factors effecting rate-performance, allowing us to assess performance or gain mechanistic insights. Clearly, such a model will never be as accurate or as comprehensive as the DFN model. However, so long as it is reasonably accurate, it will provide an extremely valuable tool for the first-order analysis of experimental data.”

Reviewer #3

This manuscript presents very useful modelling strategies for predicting the rate characteristics of batteries. Although is prior literature on this particular topic, the present manuscript presents a sufficiently novel and useful approach to warrant publication. Otherwise, the paper is well written, thorough, and the methods and results well documented. I find no issues that need to be addressed prior to publication in "Nature Communications". This will be an impactful paper for the scientific community that is working on energy-storage technologies.

Answer: Thanks very much for your very kind comments. No response required

REVIEWERS' COMMENTS:

Reviewer #2 (Remarks to the Author):

My opinion of the manuscript has not changed. I do not recommend publication.

Reviewer #3

Revisions made are sufficient to warrant publication in "Nature Communications"